# A Vision Aided Initial Alignment Method of Strapdown Inertial Navigation Systems in Polar Regions

**DOI:** 10.3390/s22134691

**Published:** 2022-06-21

**Authors:** Fubin Zhang, Xiaohua Gao, Wenbo Song

**Affiliations:** School of Marine Science and Technology, Northwestern Polytechnical University, 127 West Youyi Road, Xi’an 710072, China; zhangfubin@nwpu.edu.cn (F.Z.); songwenbo456@outlook.com (W.S.)

**Keywords:** SINS, AUV, polar region, vision

## Abstract

The failure of the traditional initial alignment algorithm for the strapdown inertial navigation system (SINS) in high latitude is a significant challenge due to the rapid convergence of polar longitude. This paper presents a novel vision aided initial alignment method for the SINS of autonomous underwater vehicles (AUV) in polar regions. In this paper, we redesign the initial alignment model by combining inertial navigation mechanization equations in a transverse coordinate system (TCS) and visual measurement information obtained from a camera fixed on the vehicle. The observability of the proposed method is analyzed under different swing models, while the extended Kalman filter is chosen as an information fusion algorithm. Simulation results show that: the proposed method can improve the accuracy of the initial alignment for SINS in polar regions, and the deviation angle has a similar estimation accuracy in the case of uniaxial, biaxial, and triaxial swing modes, which is consistent with the results of the observable analysis.

## 1. Introduction

Autonomous underwater vehicles (AUV) have always been an important tool to undertake Marine military tasks and complete Marine resource development, especially in polar resource exploration, and has attracted more and more attention from researchers around the world [1,2]. Due to the special geographical environment and geomagnetic characteristics in polar regions, common satellite navigation, radio navigation, and geomagnetic navigation can not work effectively in polar regions for a long time [3,4,5]. Ionospheric scintillation often occurs in high latitude areas, and the strong phase change and amplitude fluctuation of the signal may interfere with the work of the global positioning system (GPS) receiver, which has a great impact on the accuracy, reliability, and availability of GPS systems [6]. The inertial navigation system (INS) has high autonomy and is not affected by external factors such as climate and positions, so it can continuously provide speed and attitude information. Therefore, as one of the main components of the AUV navigation system, the inertial navigation system has become the key for AUV to complete its polar resource exploration mission [7].

However, accurate initial alignment of navigation equipment must be completed before starting navigation; otherwise, navigation accuracy will be affected [8]. However, with the increase of latitude, the included angle between the angular velocity vector of the Earth’s rotation and the gravitational acceleration vector decreases until it overlaps. Therefore, the strap-down inertial navigation system cannot achieve self-alignment in the polar region [9,10,11], and the initial alignment must be completed by other methods. Therefore, the drift navigation system [12], grid navigation system [13,14,15], and horizontal navigation system [16,17] are designed and developed, and initial alignment is carried out on this basis. However, none of these navigation systems has global navigation capability. The transfer alignment of navigation system based on the main inertial navigation system (MINS) is the main method of strap-down inertial navigation system initial alignment in the polar region [18,19]. In order to realize the initial alignment of the polar region, a fast transfer alignment scheme of speed and attitude matching is proposed in the [20], which requires the aircraft to perform simple motions. In Ref. [21], aiming at the initial alignment problem of ship inertial navigation in the polar region, an MINS transfer alignment model based on an inverse coordinate system is established. Based on the grid coordinate system, Ref. [22] estimates and corrects the inertial navigation errors of airborne weapons by matching the speed and attitude of the information of the main inertial navigation system with the polar transfer alignment method of airborne weapons. However, these algorithms are based on the information of the main inertial navigation system. These methods cannot be used to perform initial alignments of inertial navigation on an AUV without a high precision main inertial navigation system. Therefore, the application of these methods has great limitations.

In recent years, the topic of vision-assisted inertial navigation has attracted extensive attention. The low cost, low weight, and low power of the camera make it an ideal auxiliary system for the inertial navigation system, and it has certain applications in indoor auxiliary navigation [23], UAV navigation [24], and intelligent vehicle navigation [25]. After function extraction, function matching, tracking, and motion estimation of image sequence, the attitude and orientation information are updated. The visual feature points and constraints between the two images are obtained by matching the camera, so as to determine the movement information of the device [26,27]. On this basis, vision and navigation technology are gradually integrated and become a new research focus and development direction in the navigation field.

Aiming at the problem that the strapdown inertial navigation system of AUV cannot self-align in the polar region, this paper proposes a visually-assisted initial alignment method of strapdown inertial navigation by combining vision with inertial navigation and taking the visual measurement position and attitude as the observation amount. In practice, the strapdown inertial navigation system (SINS) on the AUV can use known feature points for alignment during the day and stars for alignment at night. Because it does not rely on other high precision navigation equipment, it can meet the use of complex situations.

The main work of this paper is as follows:(1)To solve the problem that strap-down inertial navigation system cannot use a traditional mechanized system in a high latitude area, the establishment of a horizontal mechanized system based on a horizontal coordinate system (TCS) can better meet the requirements of initial alignment of AUV in polar areas.(2)Designing an initial alignment method of SINS assisted by visual measurement information, combine SINS with visual measurement, update the state equation and measurement equation by using the observation result of AUV’s motion as the measurement information, design extended kalman filter (EKF), and obtain the initial alignment method of SINS assisted by vision.(3)The observability measure of the initial alignment method of the SINS was analyzed, and the initial alignment simulation experiment of the shaking base was designed. The same alignment accuracies was obtained through different swinging modes.

The rest of this article is organized as follows: Section 2 describes the establishment of the TCS and horizontal mechanized system. In Section 3, SINS is combined with visual measurement, and an EKF alignment method designed by the visual model is proposed. Section 4 constructs the observability matrix and analyzes the observability of the algorithm. In Section 5, numerical simulation and simulation of strap-down inertial navigation system in the polar region are carried out to verify the performance of the method. In Section 6, the pitch uniaxial swing scheme is used for experimental verification. Finally, the conclusions and future work are summarized in Section 7.

## 2. Polar Initial Alignment Algorithm Based on Abscissa System

### 2.1. Transversal Earth Coordinate System (TEF) and Transversal Geographic Coordinate System (TGF) Definition and Parameter Conversion

The horizontal longitude and latitude of a sphere are defined by referring to the definition method of geographic longitude and latitude of a sphere, as shown in Figure 1. The traditional longitude line is turned to the equator, which is called pseudo-longitude and latitude. The transverse earth coordinate system, namely $e_{t}$ system, uses $90^{\circ }$ E/W coils as the pseudo equator, $0^{\circ }/180^{\circ }$ coils as the pseudoprime meridians, and the intersection of the equator and the pseudoprime meridian as the pseudo pole. In the horizontal earth coordinate system, the northeast celestial geographic coordinate system is called the transverse geographic coordinate system, denoted as the $g_{t}$ system. Taking point P as an example, the direction of point P tangent to the pseudo meridian and pointing to the pseudo-North Pole is defined as false north ($oy_{gt}$), and the direction of point P perpendicular to the local horizontal plane is defined as false celestial ($oz_{gt}$). The P point is tangent to the pseudo-east coil ($ox_{gt}$). $ox_{gt}$, $oy_{gt}$, and $oz_{gt}$ constitute the right-handed Cartesian coordinate system. The $ox_{gt}y_{gt}z_{gt}$ coordinate system is the horizontal geographic coordinate system defined by the pseudo-earth coordinate system.

According to the definition of TCS, the *e*-frame can turn to the $e_{t}$-frame by rotating the *e*-frame around axis $oy_{e}$. Based on the theory of rotation, the conversion relationship between *e*-frame and $e_{t}$-frame can be expressed as follows:(1)$$C_{e}^{e_{t}}=\left[\begin{array}{ccc} \cos\left(-90^{\circ }\right) & 0 & -\sin\left(-90^{\circ }\right)\\ 0 & 1 & 0\\ \sin\left(-90^{\circ }\right) & 0 & \cos\left(-90^{\circ }\right) \end{array}\right]=\left[\begin{array}{ccc} 0 & 0 & 1\\ 0 & 1 & 0\\ -1 & 0 & 0 \end{array}\right]$$

By relying on the mathematical model of the spherical right angle, the transfer relationship of latitude and longitude between *g*-frame and $g_{t}$-frame can be defined as:(2)$$L_{t}=-\arcsin\left(\cos L\cos\lambda \right)$$
(3)$$\lambda_{t}=\arctan\frac{\cos L\sin\lambda }{\sin L_{t}}$$

In addition, the conversion relationship between the $g_{t}$-frame and *g*-frame can be expressed as follows:(4)$$C_{g}^{g_{t}}=C_{e_{t}}^{g_{t}}C_{e}^{e_{t}}C_{g}^{e}$$
where the transformation matrix of the *e*-frame to the *g*-frame is:(5)$$C_{e}^{g}=\left[\begin{array}{ccc} -\sin\lambda & \cos\lambda & 0\\ -\sin L\cos\lambda & -\sin L\sin\lambda & \cos L\\ \cos L\cos\lambda & \cos L\sin\lambda & \sin L \end{array}\right]$$
(6)$$C_{e_{t}}^{g_{t}}=\left[\begin{array}{ccc} -\sin\lambda_{t} & \cos\lambda_{t} & 0\\ -\sin L_{t}\cos\lambda_{t} & -\sin L_{t}\sin\lambda_{t} & \cos L_{t}\\ \cos L_{t}\cos\lambda_{t} & \cos L_{t}\sin\lambda_{t} & \sin L_{t} \end{array}\right]$$

### 2.2. The Mechanization Equations of Latitude and Longitude in the $e_{t}$-Frame

The transversal *g*-frame is defined as the navigation coordinate system in the transversal *n*-frame, and its mechanical equations are similar to the North-pointed INS.

(1)Attitude angle differential equation:

(7)$${\dot{C}}_{b}^{g_{t}}=C_{b}^{g_{t}}\Omega_{g_{t}b}^{b}$$
where $\Omega_{g_{t}b}^{b}$ represents the cross product of the anti-symmetric matrix of $\omega_{g_{t}b}^{b}$.
(8)$$\omega_{g_{t}b}^{b}=\omega_{ib}^{b}-C_{g_{t}}^{b}\omega_{ig_{t}}^{g_{t}}$$

(2)Velocity differential equation:

The mechanics equation of the vehicle in TNF is shown as follows:(9)$${\dot{\upsilon }}^{g_{t}}=f^{g_{t}}-\left(\omega_{eg_{t}}^{g_{t}}+2\omega_{ie}^{g_{t}}\right)\times \upsilon^{g_{t}}+g^{g_{t}}$$
where $f^{g_{t}}=C_{b}^{g_{t}}f^{b}$, $g=C_{g}^{g_{t}}g^{g}\approx {\left[\begin{array}{ccc} 0 & 0 & -g \end{array}\right]}^{T}$.

(3)Position differential equation:

The differential equation of transversal latitude $L_{t}$, transversal longitude $\lambda_{t}$, and height *h* can be expressed as follows:(10)$${\dot{L}}_{t}=\frac{\upsilon_{N_{t}}}{R_{ot}+h},{\dot{\lambda }}_{t}=\frac{\upsilon_{E_{t}}}{\left(R_{ot}+h\right)\cos L_{t}},\dot{h}=\upsilon_{U_{t}}$$

(4)Error equation of attitude:

In the case where the scale factor error and installation error of the SINS gyroscope have been compensated, the vector form of the attitude error equation can be expressed as follows:(11)$${\dot{\phi }}^{n_{t}}=\phi^{n_{t}}\times \omega_{in_{t}}^{n_{t}}+\delta \omega_{in_{t}}^{n_{t}}-C_{b}^{n_{t}}\delta \omega_{ib}^{b}$$
where the partial derivative $\delta \omega_{in_{t}}^{n_{t}}$ of $\omega_{in_{t}}^{n_{t}}$ with pseudo speed $\upsilon_{E_{t}}$, $\upsilon_{N_{t}}$, $\upsilon_{U_{t}}$ and transversal position $L_{t}$, $\lambda_{t}$, *h* is:(12)$$\delta \omega_{in_{t}}^{n_{t}}=\delta \omega_{ie_{t}}^{n_{t}}+\delta \omega_{e_{t}n_{t}}^{n_{t}}$$
(13)$$\delta \omega_{ie_{t}}^{n_{t}}=\omega_{ie}\left[\begin{array}{ccc} 0 & \cos\lambda_{t} & 0\\ -\cos L_{t}\cos\lambda_{t} & \sin L_{t}\sin\lambda_{t} & 0\\ -\sin L_{t}\cos\lambda_{t} & -\cos L_{t}\sin\lambda_{t} & 0 \end{array}\right]\left[\begin{array}{c} \delta L_{t}\\ \delta \lambda_{t}\\ \delta h \end{array}\right]$$
(14)$$\delta \omega_{e_{t}n_{t}}^{n_{t}}=\left[\begin{array}{ccc} 0 & -\frac{1}{R_{ot}+h} & 0\\ \frac{1}{R_{ot}+h} & 0 & 0\\ \frac{\tan L}{R_{ot}+h} & 0 & 0 \end{array}\right]\left[\begin{array}{c} \delta \upsilon_{E_{t}}\\ \delta \upsilon_{N_{t}}\\ \delta \upsilon_{h_{t}} \end{array}\right]+\left[\begin{array}{ccc} 0 & 0 & \frac{\upsilon_{N_{t}}}{{\left(R_{ot}+h\right)}^{2}}\\ 0 & 0 & -\frac{\upsilon_{E_{t}}}{{\left(R_{ot}+h\right)}^{2}}\\ \frac{\upsilon_{E_{t}}}{R_{ot}+h}\sec^{2}L & 0 & \frac{\upsilon_{E_{t}}\tan L_{t}}{{\left(R_{ot}+h\right)}^{2}} \end{array}\right]\left[\begin{array}{c} \delta L_{t}\\ \delta \lambda_{t}\\ \delta h \end{array}\right]$$

Gyro error $\delta \omega_{ib}^{b}$ is composed of random constant $\varepsilon_{b}$ and Gaussian white noise $\varepsilon_{w}$:(15)$$\left\{\begin{array}{c} \delta \omega_{ib}^{b}=\varepsilon_{b}+\varepsilon_{w}\\ {\dot{\varepsilon }}_{b}=0 \end{array}\right.$$

(5)Velocity error equation

In the case where the accelerometer’s scale factor error and installation error have been compensated, the vector form of the velocity error equation can be expressed as follows:(16)$$\begin{array}{c} \delta {\dot{\upsilon }}^{n_{t}}=-\phi^{n_{t}}\times C_{b}^{n_{t}}f_{b}+\delta \upsilon^{n_{t}}\times \left(2\omega_{ie_{t}}^{n_{t}}+\omega_{e_{t}n_{t}}^{n_{t}}\right)\\ \;\;\;\;\;\;\;+\upsilon^{n_{t}}\times \left(2\delta \omega_{ie_{t}}^{n_{t}}+\delta \omega_{e_{t}n_{t}}^{n_{t}}\right)+C_{b}^{n_{t}}\delta f^{b} \end{array}$$

Assume that the accelerometer error $\delta f^{b}$ consists of a random constant $\nabla_{b}$ and Gaussian white noise $\nabla_{w}$, i.e., $\left\{\begin{array}{c} \delta f^{b}=\nabla_{b}+\nabla_{w}\\ {\dot{\nabla }}_{b}=0 \end{array}\right.$.

(6)Position error equation

The component of the position error equation is:(17)$$\left[\begin{array}{c} \delta {\dot{L}}_{t}\\ \delta {\dot{\lambda }}_{t}\\ \delta \dot{h} \end{array}\right]=\left[\begin{array}{ccc} 0 & \frac{1}{R_{ot}+h} & 0\\ \frac{\sec L_{t}}{R_{ot}+h} & 0 & 0\\ 0 & 0 & 1 \end{array}\right]\left[\begin{array}{c} \delta \upsilon_{E_{t}}\\ \delta \upsilon_{N_{t}}\\ \delta \upsilon_{U_{t}} \end{array}\right]+\left[\begin{array}{ccc} 0 & 0 & \frac{\upsilon_{N_{t}}}{{\left(R_{ot}+h\right)}^{2}}\\ \frac{\upsilon_{E_{t}}\tan L_{t}\sec L_{t}}{R_{ot}+h} & 0 & -\frac{\upsilon_{E_{t}}\sec L_{t}}{{\left(R_{ot}+h\right)}^{2}}\\ 0 & 0 & 0 \end{array}\right]\left[\begin{array}{c} \delta L_{t}\\ \delta \lambda_{t}\\ \delta h \end{array}\right]$$

## 3. Visually Assisted SINS Initial Alignment Algorithm in the Polar Region

### 3.1. The Theoretical Basis of Visual Positioning

The system state variables of filter can be expressed as follows:(18)$$X={\left[\begin{array}{ccccccc} \phi^{n_{t}} & \delta V^{n_{t}} & \delta T_{}^{n_{t}} & \varepsilon^{b} & \nabla^{b} & \theta & \delta l_{bc}^{b} \end{array}\right]}^{T}$$
where $\phi^{n_{t}}$ is the misalignment angle of SINS, $\delta V^{n_{t}}$ is the velocity error, $\delta T^{n_{t}}$ is the position error, $\varepsilon^{b}$ is the constant drift error of gyroscope, $\nabla^{b}$ is the constant drift error of accelerometer, θ is the setting-angle error between INS and camera. The error caused by lever arm effect between camera and INS is $\delta l_{bc}^{b}$.

The state equation of the system can be expressed as follows:(19)$$\left\{\begin{array}{c} {\dot{\phi }}^{n_{t}}=\phi^{n_{t}}\times \omega_{in_{t}}^{e}+\delta \omega_{in_{t}}^{n_{t}}-C_{b}^{n_{t}}\delta \omega_{ib}^{b}\\ \delta {\dot{\upsilon }}^{n_{t}}=-\phi^{n_{t}}\times C_{b}^{n_{t}}f_{b}+\delta \upsilon^{n_{t}}\times \left(2\omega_{ie_{t}}^{n_{t}}+\omega_{e_{t}n_{t}}^{n_{t}}\right)+\upsilon^{n_{t}}\times \left(2\delta \omega_{ie_{t}}^{n_{t}}+\delta \omega_{e_{t}n_{t}}^{n_{t}}\right)+C_{b}^{n_{t}}\delta f^{b}\\ \delta {\dot{T}}^{n_{t}}=\delta V^{n_{t}}\\ {\dot{\varepsilon }}_{b}^{b}=0\\ {\dot{\nabla }}_{b}^{b}=0\\ \dot{\theta }=0\\ \delta {\dot{l}}_{bc}^{b}=0 \end{array}\right.$$

$n_{t}$ is the navigation coordinate system, and the horizontal geographic coordinate system $g_{t}$ is the navigation coordinate system $n_{t}$.

### 3.2. The Establishment of the Measurement Equation of Filter

The transformation relation between AUV body coordinate system and camera coordinate system (c) is represented by a 3×3 dimensional rotation matrix *C* and a translation vector *T*:(20)$$\left[\begin{array}{c} X_{f}\\ Y_{f}\\ Z_{f} \end{array}\right]=C\left[\begin{array}{c} X_{C}\\ Y_{C}\\ Z_{C} \end{array}\right]+T$$

The above formula is expressed in the homogeneous coordinate form:(21)$$\left[\begin{array}{c} X_{f}\\ Y_{f}\\ Z_{f}\\ 1 \end{array}\right]=\left[C/T\right]\left[\begin{array}{c} X_{C}\\ Y_{C}\\ Z_{C}\\ 1 \end{array}\right]=M\left[\begin{array}{c} X_{C}\\ Y_{C}\\ Z_{C}\\ 1 \end{array}\right]=\left[\begin{array}{cccc} C_{11} & C_{12} & C_{13} & T_{14}\\ C_{21} & C_{22} & C_{23} & T_{24}\\ C_{31} & C_{32} & C_{33} & T_{34}\\ 0 & 0 & 0 & 1 \end{array}\right]\left[\begin{array}{c} X_{C}\\ Y_{C}\\ Z_{C}\\ 1 \end{array}\right]$$
where *M* is defined as the relative pose transfer matrix, where:(22)$$\left\{\begin{array}{c} T_{14}=-C_{11}X_{f}-C_{12}Y_{f}-C_{13}Z_{f}\\ T_{24}=-C_{21}X_{f}-C_{22}Y_{f}-C_{23}Z_{f}\\ T_{34}=-C_{31}X_{f}-C_{32}Y_{f}-C_{33}Z_{f} \end{array}\right.$$
where $T_{11}-T_{33}$ is the component of the rotation matrix *T* of the AUV body coordinate system and camera coordinate system.

According to the pinhole imaging model, the conversion relationship between the camera coordinate system and image plane coordinate system is as follows:(23)$$Z_{C}\left[\begin{array}{c} x_{p}\\ y_{p}\\ 1 \end{array}\right]=\left[\begin{array}{cccc} f & 0 & 0 & 0\\ 0 & f & 0 & 0\\ 0 & 0 & 1 & 0 \end{array}\right]\left[\begin{array}{c} X_{C}\\ Y_{C}\\ Z_{C}\\ 1 \end{array}\right]$$
where *f* is the focal length of the camera. Substitute Equation (20) into Equation (23), and derive:(24)$$Z_{C}\left[\begin{array}{c} u\\ v\\ 1 \end{array}\right]=\left[\begin{array}{ccc} 1/d_{x} & 0 & u_{0}\\ 0 & 1/d_{y} & v_{0}\\ 0 & 0 & 1 \end{array}\right]\left[\begin{array}{cccc} f & 0 & 0 & 0\\ 0 & f & 0 & 0\\ 0 & 0 & 1 & 0 \end{array}\right]\left[C/T\right]\left[\begin{array}{c} X_{C}\\ Y_{C}\\ Z_{C}\\ 1 \end{array}\right]=M_{i}M_{e}\left[\begin{array}{c} X_{C}\\ Y_{C}\\ Z_{C}\\ 1 \end{array}\right]$$

In the formula, the internal parameter matrix $M_{i}$ contains the parameters $f,d_{x},d_{y},u_{0},v_{0}$ that reflect the internal optical and geometric characteristics of the camera, and the external parameter matrix $M_{e}$ contains the attitude transfer matrix *C* and position transfer vector *T* that reflects the spatial position relationship between the camera coordinate system and the three-dimensional reference coordinate system.

(1)Visual measurement equation

The position coordinates of feature points in the image plane coordinate system (*P* system) can be obtained from the image. The subscript *i* indicates the *i*th feature point.

The feature points satisfy the collinear equation:(25)$$Z_{i}^{c}\left[\begin{array}{c} u_{i}\\ v_{i} \end{array}\right]=\left[\begin{array}{ccc} f/d_{x} & 0 & u_{0}\\ 0 & f/d_{y} & v_{0} \end{array}\right]C_{n_{t}}^{c}\left(T_{i}^{n_{t}}-T_{o_{c}}^{n_{t}}\right)$$

$T_{i}^{n_{t}}$ is the position coordinate of the feature point *i* in the *n*-frame. $T_{o_{c}}^{n_{t}}$ is the position coordinate of camera optical center under the *n*-frame. $C_{n_{t}}^{c}$ is the attitude transformation matrix of the $n_{t}$-frame to the *c*-frame. ${\left[u_{i}\;v_{i}\right]}^{T}$ is the position coordinate of the imaging of feature point *i* in the *p*-frame. $Z_{i}^{c}$ is the projection length of the distance from the feature point *i* to the optical center of the camera on the optical axis. $f/d_{x}$ and $f/d_{y}$ are the camera’s equivalent focal lengths. $u_{0}$ and $v_{0}$ are the position coordinates of the intersection of the camera optical axis and the image in the *p*-frame. For the camera that has been calibrated, $f,d_{x},d_{y},u_{0},v_{0}$ are known quantities.

The position coordinates of the image of *n* (*n* > 6) feature points in the *p*-frame and the position coordinates of feature points in the *n*-frame can be obtained at time *t*. Based on the linear approximation of the position coordinates of feature points in the *p*-frame and *n*-frame, the transformation matrix $C_{n_{t}}^{c}\left(t\right)$ of the *n*-frame to the *c*-frame at this moment and the position $T^{n_{t}}\left(t\right)$ of the camera under the navigation system can be obtained by using the principle of the least-squares method.

The relationship between the transformation matrix $C_{n_{t}}^{c}\left(t\right)$ and $C_{b}^{n_{t}}\left(t\right)$ can be expressed as follows:(26)$$C_{b}^{n_{t}}\left(t\right)={\left[{\left(C_{b}^{c}\right)}^{-1}C_{n_{t}}^{c}\left(t\right)\right]}^{T}$$

The relationship between the position $T^{n_{t}}\left(t\right)$ and $T_{b}^{n_{t}}\left(t\right)$ can be expressed as follows:(27)$$T_{b}^{n_{t}}\left(t\right)=T_{}^{n_{t}}\left(t\right)-C_{b}^{n_{t}}\left(t\right)l_{bc}^{b}$$
where the $C_{b}^{n_{t}}\left(t\right)$ is the transformation matrix from *b*-frame to $n_{t}$-frame, the $T_{b}^{n_{t}}\left(t\right)$ is the position coordinates of the vehicle under the *n*-frame.

After the camera is calibrated and fixed with the vehicle, the transformation matrix $C_{b}^{c}$ and the translation vector $l_{bc}^{b}$ of the *b*-frame to the *c*-frame can be obtained. Then, the attitude transformation matrix $C_{b}^{n_{t}}\left(t\right)$ and the position coordinate $T_{b}^{n_{t}}\left(t\right)$ of the vehicle under the *n*-frame can be calculated with $C_{n_{t}}^{c}\left(t\right)$ and $T^{n_{t}}\left(t\right)$.

(2)Establishment of the measurement equation

The measurement equation of the system can be expressed as follows:(28)$$z=\left[\begin{array}{c} Mat2Ang\left\{{\tilde{C}}_{c}^{n_{t}}{\tilde{C}}_{b}^{c}{\tilde{C}}_{n_{t}}^{b}\right\}\\ T_{b}^{n_{t}}\left(t\right)+C_{b}^{n_{t}}\left(t\right)l_{bc}^{b}-T_{c}^{n_{t}}\left(t\right) \end{array}\right]\approx \left[\begin{array}{c} -\phi^{n_{t}}+C_{b}^{n_{t}}{\left(C_{b}^{c}\right)}^{T}\theta \\ \delta T^{n_{t}}\left(t\right)+C_{b}^{n_{t}}\delta l_{bc}^{b} \end{array}\right]+\omega_{c}$$
where Mat2Ang{} is the attitude angle corresponding to the attitude transformation matrix. ${\tilde{C}}_{c}^{n_{t}}$ is the attitude matrix measured by the camera. ${\tilde{C}}_{b}^{c}$ is the transformation matrix between camera to INS. ${\tilde{C}}_{n_{t}}^{b}$ is the attitude matrix measured by the INS. $T_{b}^{n_{t}}\left(t\right)$ is the position measured by the INS. $T_{c}^{n_{t}}\left(t\right)$ is the position measured by the camera. $\omega_{c}$ is the visual measurement noise. $C_{b}^{c}$ is the transformation matrix of the *b*-frame to the *c*-frame. Thereinto, the schematic diagram of visual aided INS alignment platform is shown in Figure 2.

## 4. Observability Analysis of SINS Polar Initial Alignment Algorithm with Visual Assistance

The observability of the system is different under different swaying modes. Observability analysis theory is used to analyze the observability of the system under different maneuvers and to find the maneuvering method that is most suitable for AUV initial alignment in the polar region.

In this paper, we analyze the observability of the system by the analytical method.

When swaying around the inertial measurement unit (IMU), since the position of the IMU does not change and the true velocity is zero, the error equation can be simplified to make the analysis intuitive and simple. Then, we can get:(29)$${\dot{\phi }}^{n_{t}}=-\omega_{in_{t}}^{e}\times \phi^{n_{t}}-\xi^{n}$$
(30)$$\delta {\dot{\upsilon }}^{n_{t}}=-\phi^{n_{t}}\times C_{b}^{n_{t}}f_{b}+\nabla^{n}$$
(31)$$\delta {\dot{T}}^{n_{t}}=\delta V^{n_{t}}$$

The state space model is:(32)$$\left\{\begin{array}{c} \dot{X}=AX+GW\\ Z=HX+V \end{array}\right.$$
where:(33)$$A=\left[\begin{array}{c} \begin{array}{ccccccc} w_{ie}\times & 0_{33} & 0_{33} & C_{b}^{n} & 0_{33} & 0_{33} & 0_{33}\\ g\times & 0_{33} & 0_{33} & 0_{33} & C_{b}^{n} & 0_{33} & 0_{33}\\ 0_{33} & I_{33} & 0_{33} & 0_{33} & 0_{33} & 0_{33} & 0_{33} \end{array}\\ 0_{\left(12,3\right)} \end{array}\right]$$
(34)$$H=\left[\begin{array}{ccccccc} I_{33} & 0_{33} & 0_{33} & 0_{33} & 0_{33} & C_{b}^{n} & 0_{33}\\ 0_{33} & 0_{33} & I_{33} & 0_{33} & 0_{33} & 0_{33} & C_{b}^{n} \end{array}\right]$$

$Q=\left[H^{T}\;{\left(HA\right)}^{T}\;\cdots \;{\left(HA^{20}\right)}^{T}\right]$ is observability matrix, $Z={\left[y^{T},{\dot{y}}^{T},{\cdots ,(y^{\left(20\right)})}^{T}\right]}^{T}$ is the matrix consisting of observables and their derivatives. We use rows 1th to 18th for analysis:(35)$$\left\{\begin{array}{c} {\tilde{\phi }}_{E}=\phi_{E}+C_{11}\theta_{\mathrm{right}}+C_{12}\theta_{front}+C_{13}\theta_{up}\\ {\tilde{\phi }}_{N}=\phi_{N}+C_{21}\theta_{right}+C_{22}\theta_{front}+C_{23}\theta_{up}\\ {\tilde{\phi }}_{U}=\phi_{U}+C_{31}\theta_{right}+C_{32}\theta_{front}+C_{33}\theta_{up}\\ {\tilde{T}}_{E}=T_{E}+C_{11}l_{right}+C_{12}l_{front}+C_{13}l_{up}\\ {\tilde{T}}_{N}=T_{N}+C_{21}l_{right}+C_{22}l_{front}+C_{23}l_{up}\\ {\tilde{T}}_{U}=T_{U}+C_{31}l_{right}+C_{32}l_{front}+C_{33}l_{up}\\ {\dot{\tilde{\phi }}}_{E}=w_{u}\phi_{N}-w_{N}\phi_{U}+\xi_{E}^{n}\\ {\dot{\tilde{\phi }}}_{N}=-w_{u}\phi_{E}-w_{E}\phi_{U}+\xi_{N}^{n}\\ {\dot{\tilde{\phi }}}_{U}=w_{u}\phi_{N}+w_{E}\phi_{N}+\xi_{U}^{n}\\ {\dot{\tilde{T}}}_{E}=\delta V_{E}\\ {\dot{\tilde{T}}}_{N}=\delta V_{N}\\ {\dot{\tilde{T}}}_{U}=\delta V_{U}\\ {\ddot{\tilde{\phi }}}_{E}=\left(-w_{U}^{2}-w_{N}^{2}\right)\phi_{E}+w_{N}w_{E}\phi_{N}+w_{U}w_{E}\phi_{U}+w_{U}\xi_{N}^{n}+w_{N}\xi_{U}^{n}\\ {\ddot{\tilde{\phi }}}_{N}=w_{N}w_{E}\phi_{E}+\left(-w_{U}^{2}-w_{E}^{2}\right)\phi_{N}+w_{U}w_{N}\phi_{U}+w_{U}\xi_{E}^{n}+w_{E}\xi_{U}^{n}\\ {\ddot{\tilde{\phi }}}_{U}=w_{U}w_{E}\phi_{E}+w_{U}w_{N}\phi_{N}+\left(-w_{N}^{2}-w_{E}^{2}\right)\phi_{U}+w_{E}\xi_{N}^{n}+w_{N}\xi_{E}^{n}\\ {\ddot{\tilde{T}}}_{E}=-g\phi_{N}+\nabla_{E}^{n}\\ {\ddot{\tilde{T}}}_{N}=g\phi_{E}+\nabla_{N}^{n}\\ {\ddot{\tilde{T}}}_{U}=\nabla_{U}^{n} \end{array}\right.$$
where:(36)$$C_{b}^{n}=\left[\begin{array}{ccc} \cos\varphi \cos\gamma -\sin\varphi \sin\eta \sin\gamma & -\sin\varphi \cos\eta & \cos\varphi \sin\gamma +\sin\varphi \sin\eta \cos\gamma \\ \sin\varphi \cos\gamma +\cos\varphi \sin\eta \sin\gamma & \cos\varphi \cos\eta & \sin\varphi \sin\gamma -\cos\varphi \sin\eta \cos\gamma \\ -\cos\eta \sin\gamma & \sin\eta & \cos\eta \cos\gamma \end{array}\right]$$

Can be written as:(37)$$C_{b}^{n}=\left[\begin{array}{ccc} C_{11} & C_{12} & C_{13}\\ C_{21} & C_{22} & C_{23}\\ C_{31} & C_{32} & C_{33} \end{array}\right]$$
(38)$$-w_{ie}^{e}\times =\left[\begin{array}{ccc} 0 & w_{U} & -w_{N}\\ -w_{U} & 0 & w_{E}\\ w_{N} & -w_{E} & 0 \end{array}\right]$$
(39)$$\xi_{E}^{n}=C_{11}\xi_{right}+C_{12}\xi_{front}+C_{13}\xi_{up}$$
(40)$$\xi_{N}^{n}=C_{21}\xi_{right}+C_{22}\xi_{front}+C_{23}\xi_{\mathrm{up}}$$
(41)$$\xi_{U}^{n}=C_{31}\xi_{right}+C_{32}\xi_{front}+C_{33}\xi_{up}$$
(42)$$\nabla_{E}^{n}=C_{11}\nabla_{right}+C_{12}\nabla_{front}+C_{13}\nabla_{up}$$
(43)$$\nabla_{N}^{n}=C_{21}\nabla_{right}+C_{22}\nabla_{front}+C_{23}\nabla_{up}$$
(44)$$\nabla_{U}^{n}=C_{31}\nabla_{right}+C_{32}\nabla_{front}+C_{33}\nabla_{up}$$

In the formula, φ is yaw, γ is roll, and η is pitch. From the order of the derivative of observations, $\phi_{E},\phi_{N},\phi_{U},\theta_{right},\theta_{front},\theta_{up},T_{E},T_{N},T_{U},l_{right},l_{front},l_{up}$ corresponds to the zero-order derivative of the observations, so they have the highest degree of observability. $\delta V_{E},\delta V_{N},\delta V_{U}$ and $\xi_{right},\xi_{front},\xi_{up}$ correspond to the first-order derivatives of the observations, and they have the second highest degree of observability. $\nabla_{right},\nabla_{front},\nabla_{up}$ corresponds to the second-order derivative of the observation with the lowest observability.

It can be seen from the first three rows that $C_{b}^{n}$ is the coefficient of $\theta_{right},\theta_{front},\theta_{up}$. $C_{b}^{n}$ is a constant in the static state, so $\theta_{right},\theta_{front},\theta_{up}$ cannot be accurately estimated by these three equations. Due to the coupling between $\phi_{E},\phi_{N},\phi_{U}$ and $\theta_{right},\theta_{front},\theta_{up}$, the inaccuracy of $\theta_{right},\theta_{front},\theta_{up}$ will affect the estimation accuracy of $\phi_{E},\phi_{N},\phi_{U}$. When in the three-axis swing, $C_{b}^{n}$ keeps changing and the coefficient of $\theta_{right},\theta_{front},\theta_{up}$ also changes. We can obtain many sets of nonlinear equations through multiple measurements at different times, and then accurately estimate the six values of $\phi_{E},\phi_{N},\phi_{U},\theta_{right},\theta_{front},\theta_{up}$. In the same way, the estimated value of $T_{E},T_{N},T_{U},l_{right},l_{front},l_{up}$ will be more accurate in the triaxial swing state than in the stationary state. In addition, $\phi_{N}=-\left({\ddot{\tilde{T}}}_{E}-\nabla_{E}^{n}\right)/g,\phi_{E}=\left({\ddot{\tilde{T}}}_{N}-\nabla_{N}^{n}\right)/g$ can be derived from the 16th and 17th rows. $\phi_{E},\phi_{N}$ can be estimated through the second derivative of the position error and the estimation accuracy is ∇/g. Therefore, the $\phi_{E}$ and the $\phi_{N}$ will converge faster and more accurately than the $\phi_{U}$ in theory. In addition, taking into account the coupling relationship between $\phi_{E},\phi_{N},\phi_{U},\theta_{right},\theta_{front},\theta_{up}$, it will have an advantage on the estimation of $\theta_{right},\theta_{front},\theta_{up}$.

However, in engineering, it is difficult to perform triaxial sway, so the following is to analyze the observability of uniaxial or biaxial sway.

For uniaxial sway, the change of pitch will affect nine values of $C_{b}^{n}$. The change of roll and yaw will affect six values of $C_{b}^{n}$. Therefore, the estimation accuracy of uniaxial sway is pitch > roll = yaw. However, the estimation accuracy will be affected by the initial attitude with uniaxial sway. One degree of freedom will be ignored at a certain initial attitude. For example, when the initial pitch and yaw are both zero, the change in yaw will only cause the change of four values of $C_{b}^{n}$. When taking biaxial sway, the initial attitude will not affect the estimation accuracy and at least eight values in $C_{b}^{n}$ will change. Thus, the effects of biaxial sway and triaxial sway are basically the same in theory.

## 5. Simulation

Assume that, in the simulation experiment, the initial position of AUV in *g*-frame is $\left(89^{\circ }\;N,108^{\circ }\;E,0\;m\right)$, and in the $g_{t}$-frame is $\left(0.30^{\circ }\;N,0.95^{\circ }\;E,0\;m\right)$. The initial state of yaw is $45^{\circ }$, pitch and rolls are $10^{\circ }$, and the speed is 0m/s. AUV sways around the yaw axis, pitch axis, and roll axis with a swing amplitude of $\pm 15^{\circ }$.

Set the initial value of filter: the initial value of system state is $X\left(0\right)=0$. Initial misalignment angles are $\phi_{E}=\phi_{N}=1^{\circ }$ and $\phi_{U}=5^{\circ }$. Speed error is $\left(0.1\;m/s,0.1\;m/s,0.1\;m/s\right)$. Position error is $\left(1\;m,1\;m,1\;m\right)$. Gyro constant drift error is $0.05\left({}^{\circ }\right)/h$. Random noise is $0.01\left({}^{\circ }\right)/\sqrt{h}$. Accelerometer constant bias is 100μg. Random noise is $50\;\mu g/\sqrt{\mathrm{Hz}}$. The difference in attitude angle between the camera and the IMU is $\left(2^{\circ },5^{\circ },2^{\circ }\right)$. The misalignment angle after calibration is $\left(0.5^{\circ },0.5^{\circ },0.5^{\circ }\right)$. The lever arm between the camera and IMU is $\left(0.4\;m,1\;m,0.6\;m\right)$. The error caused by the lever arm is (0.1m,0.1m,0.1m). The accuracy of visual measurement is related to the distance between the camera and the marker. It is also related to the distance of the feature point on the marker. If the distance between the feature points is farther, the measurement accuracy will be higher. In addition, the closer the distance between the camera and the marker, the higher the measurement accuracy will be. When the distance between the feature points is 30 mm and the distance between the camera and the marker is 2000 mm, the distance error is below 10 mm, the attitude error is below $0.2^{\circ }$, and both are white noise. Thus, in this simulation, the attitude measurement noise is set to $\left(0.2^{\circ },0.2^{\circ },0.2^{\circ }\right)$, the position measurement noise is $\left(0.01\;m,0.01\;m,0.01\;m\right)$, the update period of the inertial navigation data are 20ms, the update period of the camera data are 100ms, and the filtering period is 1 s. Then, the initial variance matrix P(0), the system matrix *Q*, and the measurement noise matrix *R* are as follows:(45)$$\begin{array}{c} P\left(0\right)=20*diag\left\{{\left(1^{\circ }\right)}^{2},{\left(1^{\circ }\right)}^{2},{\left(5^{\circ }\right)}^{2},{\left(0.1\;m/s\right)}^{2},{\left(0.1\;m/s\right)}^{2},\right.{\left(0.1\;m/s\right)}^{2},\\ {\left(1\;m\right)}^{2},{\left(1\;m\right)}^{2},{\left(1\;m\right)}^{2},{\left(0.05^{\circ }/h\right)}^{2},{\left(0.05^{\circ }/h\right)}^{2},{\left(0.05^{\circ }/h\right)}^{2},{\left(100\;\mu g\right)}^{2},\\ {\left(100\;\mu g\right)}^{2},{\left(100\;\mu g\right)}^{2},{\left(0.5^{\circ }\right)}^{2},\left.{\left(0.5^{\circ }\right)}^{2},{\left(0.5^{\circ }\right)}^{2},{\left(1\;m\right)}^{2},{\left(1\;m\right)}^{2},{\left(1\;m\right)}^{2}\right\}*20 \end{array}$$
(46)$$\begin{array}{c} Q=diag\left\{{\left(0.01^{\circ }/\sqrt{h}\right)}^{2},{\left(0.01^{\circ }/\sqrt{h}\right)}^{2},{\left(0.01^{\circ }/\sqrt{h}\right)}^{2},{\left(50\;\mu g/\sqrt{\mathrm{Hz}}\right)}^{2},\right.\\ \left.{\left(50\;\mu g/\sqrt{\mathrm{Hz}}\right)}^{2},{\left(50\;\mu g/\sqrt{\mathrm{Hz}}\right)}^{2},0,0,0,0,0,0,0,0,0,0,0,0,0,0,0\right\} \end{array}$$
(47)$$R=diag\left\{{\left(0.2^{\circ }\right)}^{2},{\left(0.2^{\circ }\right)}^{2},{\left(0.2^{\circ }\right)}^{2},{\left(0.01\;m\right)}^{2},{\left(0.01\;m\right)}^{2},{\left(0.01\;m\right)}^{2}\right\}$$

According to the above simulation conditions, the Kalman filtering method is used to carry out the moving base alignment simulation with the simulation time of 200 s.

According to the above simulation conditions, The difference between the true value and the estimated value is shown in Figure 3, Figure 4 and Figure 5. The residual mounting misalignment angle between the camera and the IMU after calibration is shown in Figure 6. The residual rod arm error between the camera and IMU after calibration is shown in Figure 7.

The figure shows the state estimation under the condition of triaxial sway. Because the initial attitude error is large, in order to obtain a better estimation effect, the initial navigation system will be modified once with the current estimation value to reduce the nonlinearity of the model in the 30th second.

The estimates for the triaxial swing are shown in Table 1.

As can be seen from the table, there is little difference in the estimation accuracy of the east misalignment angle, north misalignment angle, velocity error, and position error, whether swinging or stationary. However, there is a significant difference in the estimation accuracy of sky error angle under different swinging modes. The estimation accuracy of the celestial misalignment angle is significantly higher than that of the single-axis roll, single-axis yaw, and stationary rotation when the rotation is three-axis, two-axis, and single-axis pitching. Meanwhile, the residual installation error angle between the camera and the inertial navigation can be accurately estimated, which is consistent with the theoretical analysis. When the gyroscope is biased at $0.05^{\circ }/h$ and the accelerometer is biased at 100μg, the triaxial alignment accuracy of the strap-down inertial navigation system is $0.11^{\prime },0.17^{\prime }$ and $-0.30^{\prime }$. For the estimation of the deviation angle, it has the same effect in the case of uniaxial, biaxial, and triaxial swing after alignment 200 s, which is consistent with the results of the observable analysis. In addition, the residual rod arm error between the camera and the inertial navigation device can only be accurately estimated under the condition of triaxial and biaxial swing, and the residual rod arm error between the camera and the inertial navigation device can not be accurately estimated under the condition of uniaxial and static swing. At the same time, the added zero drift can not be observed under any circumstances and the gyro zero bias estimation is not accurate; there is a large deviation.

In engineering, the rod arm between the camera and the inertial navigation device can be obtained through calibration, and the accuracy can be up to a millimeter-level. Therefore, the AUV can only carry out the pitching single-axis swing for initial alignment in the polar region, which will greatly reduce the difficulty of engineering alignment compared with the three-axis swing.

## 6. Experimental Verification

In Section 5, the initial alignment schemes of SINS under different maneuvering schemes are analyzed and simulated. The results show that the effect of single axis swing scheme in pitch is basically the same as that of the three-axis swing scheme. However, in practical engineering applications, it is difficult to realize the three-axis swing, and the pitching single-axis swing scheme is simple and easy to operate. Therefore, the pitch uniaxial swing scheme was adopted in this experiment to verify the initial alignment algorithm of the strapdown inertial navigation pole region based on visual assistance. Considering the feasibility and operability of practical engineering, a hand-pushed turntable is used to simulate the rocking motion of the vehicle.

STIM300 is selected as inertial measurement element, camera selection industrial camera DYSMT205A. By fixing the bracket, the camera and STIM300 are fixed on the turntable bracket to ensure that the relative position and pose of the inertial measurement element and the camera remain unchanged, as shown in Figure 8. The camera uses an AR Marker to obtain pose information. AR Marker is placed directly in front of the camera and will not move or change during the whole experiment. Before the motion of hand push simulated vehicle swing, the AR Marker position and attitude information of AR Marker were measured and recorded with high-precision equipment. The lever arm error of the system is very small and can be ignored.

A total of three groups of data were recorded in the experiment. Because of the high accuracy of pitch angle and roll angle, they are not given here. Using the vision-aided initial alignment algorithm, the yaw angle converges to $29.4^{\circ }$–$29.8^{\circ }$ after stationary. The error results with the yaw angle value obtained by the corresponding turntable are recorded in Table 2.

It can be seen from Table 2 that the yaw angle error can be limited to about $0.5478^{\circ }$, which includes the small angle error of axis unalignment between the camera and the turntable as well as the small angle error of axis unalignment between STIM300 and the turntable. The results show that the error of yaw angle is within an acceptable range and can meet the actual needs, which verifies the initial alignment algorithm of SINS based on visual assistance.

## 7. Conclusions

This paper solves the problem that the strap-down inertial navigation system of AUV cannot realize self-alignment in the polar region. Combining navigation with vision, a visually-assisted initial alignment method of SINS based on an abscise-coordinate system is proposed. When the AUV is swinging, the position and attitude of the AUV itself are calculated by the motion of known feature points relative to the camera. SINS is combined with visual measurement, and the observation result of AUV’s motion is used as the measurement information to update the state equation and measurement equation, in order to complete the high-precision initial alignment of SINS with visual assistance. Simulation results show that this algorithm can estimate the error angle effectively; the error between the estimated value and the true value is close to zero and does not diverge with time and has good convergence. The experimental results show that the yaw angle error can meet the practical needs and has strong practical application. At the same time, AUV only needs to pitch a single axis swing that can meet the needs of polar alignment, greatly reducing the difficulty of engineering to achieve the initial alignment of the AUV polar region. Therefore, the algorithm can better meet the requirements of AUV initial alignment in the polar region. Compared with the traditional polar alignment method, this method is simpler and more widely applicable. At present, only simulation verification and analysis have been carried out, and field investigation and test have been actively carried out in the later stage.

## Figures and Tables

**Figure 1 sensors-22-04691-f001:**
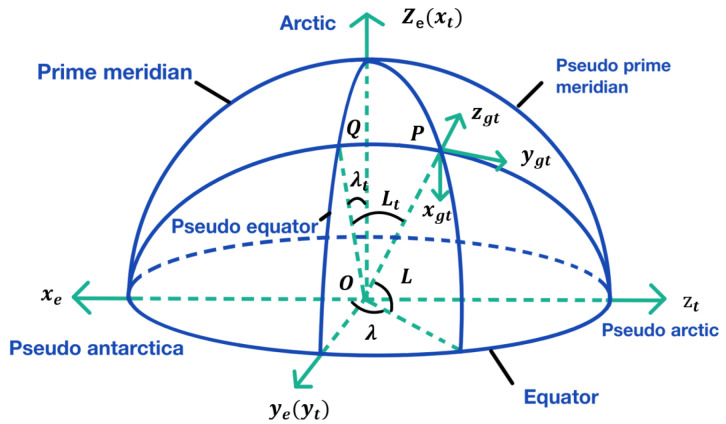
Transversal coordinate system.

**Figure 2 sensors-22-04691-f002:**
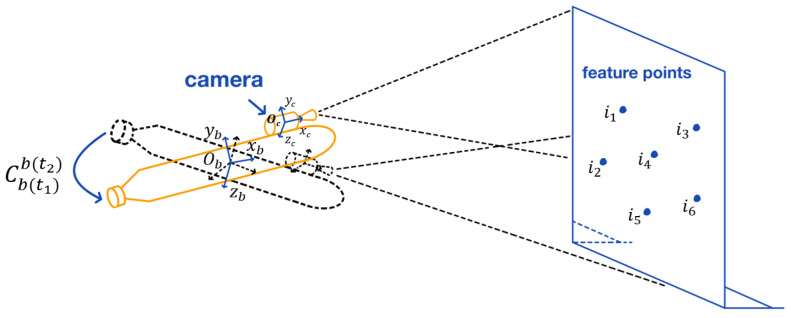
Schematic diagram of the alignment platform of the visual aided INS.

**Figure 3 sensors-22-04691-f003:**
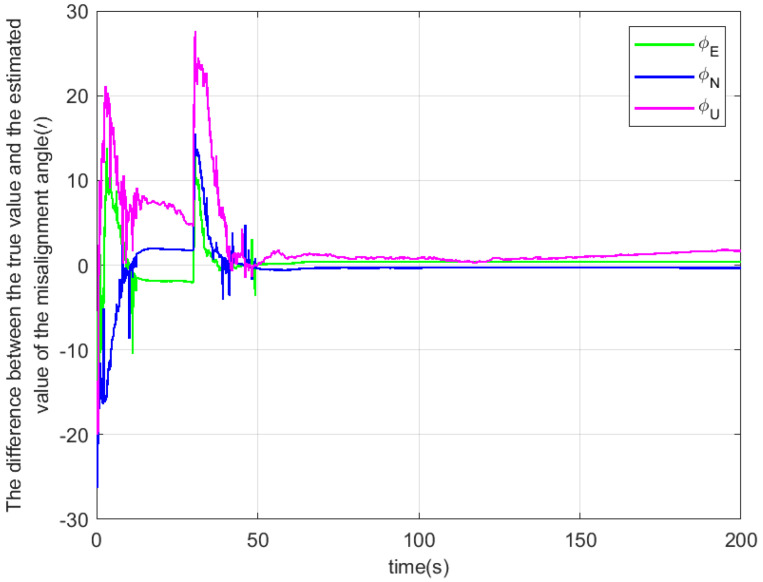
The difference between the true value and the estimated value of the misalignment angle.

**Figure 4 sensors-22-04691-f004:**
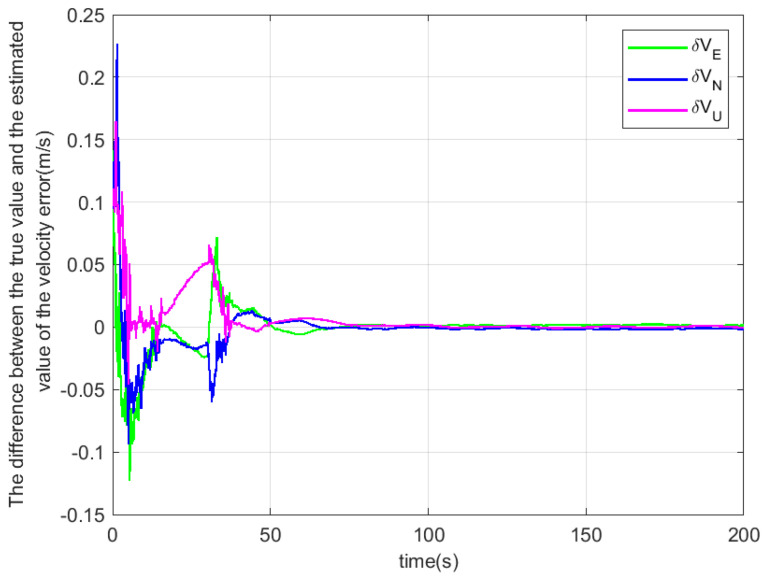
The difference between the true value and the estimated value of the velocity error.

**Figure 5 sensors-22-04691-f005:**
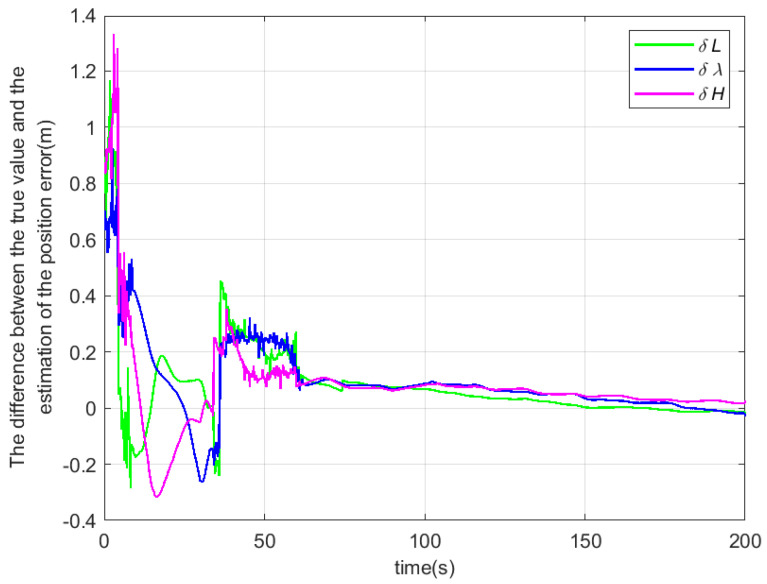
The difference between the true value and the estimated value of the position error.

**Figure 6 sensors-22-04691-f006:**
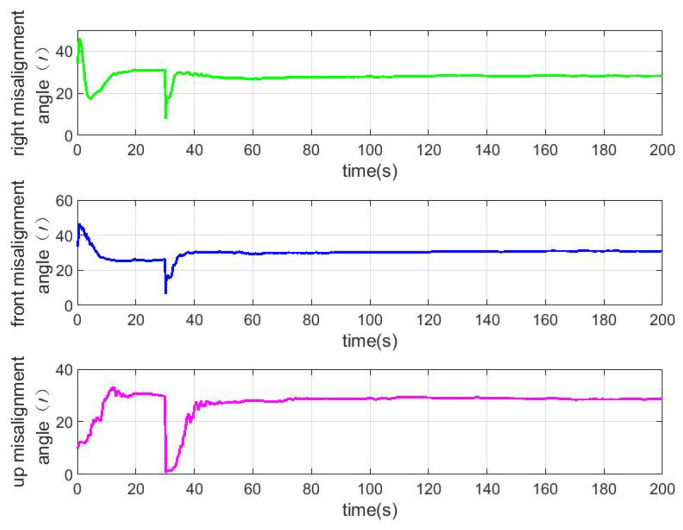
Residual mounting misalignment angle between camera and IMU after calibration.

**Figure 7 sensors-22-04691-f007:**
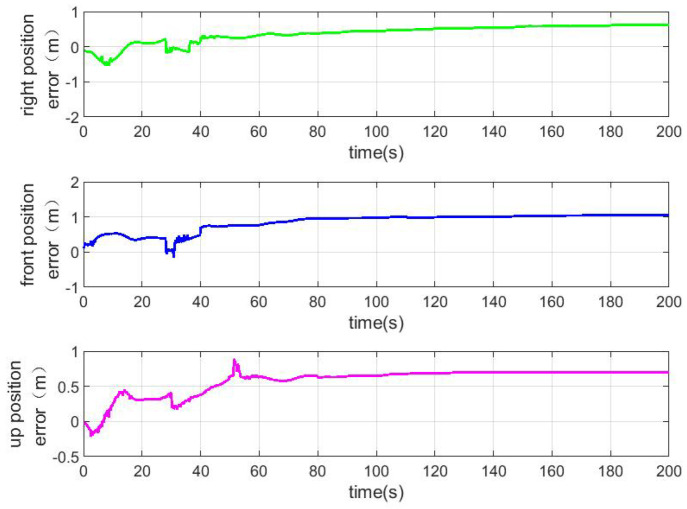
Residual bar arm error between camera and IMU after calibration.

**Figure 8 sensors-22-04691-f008:**
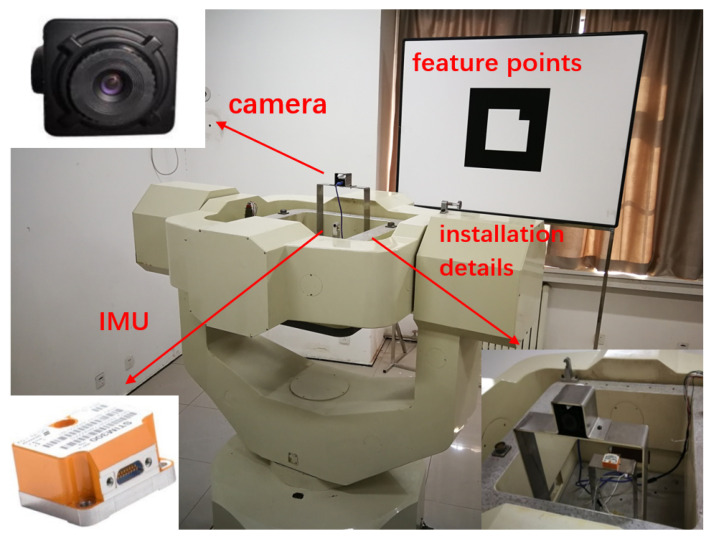
Overall connection diagram of the experiment.

**Table 1 sensors-22-04691-t001:** Estimates under triaxial sway.

Rotation Axis	$x_{b}$,$y_{b}$,$z_{b}$	$x_{b}$,$z_{b}$	$x_{b}$,$y_{b}$	$y_{b}$,$z_{b}$	$z_{b}$	$x_{b}$	$y_{b}$	Still
$\phi_{E}\;{(}^{\prime })$	0.11	0.26	0.61	0.41	0.50	0.40	0.36	0.51
$\phi_{N}\;{(}^{\prime })$	−0.17	−0.12	0.31	0.70	−0.11	−0.21	0.37	−0.18
$\phi_{U}\;{(}^{\prime })$	−0.30	−0.75	0.90	0.76	0.86	−3.20	9.21	6.38
$\delta v_{E}\;\left(m/s\right)$	6.98 $\times \;10^{-3}$	−1.20 $\times \;10^{-3}$	−3.0 $\times \;10^{-3}$	−1.20 $\times \;10^{-3}$	1.4 $\times \;10^{-3}$	−1.2 $\times \;10^{-3}$	−6.64 $\times \;10^{-4}$	−8.67 $\times \;10^{-4}$
$\delta v_{N}\;\left(m/s\right)$	8.71 $\times \;10^{-3}$	1.1 $\times \;10^{-3}$	−2.10 $\times \;10^{-3}$	−1.24 $\times \;10^{-3}$	5.4 $\times \;10^{-4}$	1.0 $\times \;10^{-3}$	−5.57 $\times \;10^{-4}$	−9.41 $\times \;10^{-4}$
$\delta v_{U}\;\left(m/s\right)$	3.13 $\times \;10^{-4}$	1.1 $\times \;10^{-3}$	−2.0 $\times \;10^{-3}$	−1.80 $\times \;10^{-3}$	1.3 $\times \;10^{-4}$	1.6 $\times \;10^{-3}$	−1.0 $\times \;10^{-4}$	2.68 $\times \;10^{-4}$
δL(m)	−0.007	0.022	0.035	−0.0025	−0.017	0.351	0.001	0.333
δλ(m)	−0.013	0.028	0.023	0.034	0.304	−0.012	0.001	0.323
δH(m)	0.017	0.015	0.021	0.023	−0.002	0.013	0.317	0.315
$\varepsilon_{right}\;{(}^{\circ }/h)$	−0.069	0.064	0.054	0.049	0.058	0.073	0.071	0.062
$\varepsilon_{front}\;{(}^{\circ }/h)$	0.042	0.035	0.050	0.058	0.043	0.078	0.021	0.043
$\varepsilon_{up}\;{(}^{\circ }/h)$	0.023	0.021	0.032	0.012	0.034	0.018	−0.029	0.005
$\nabla_{right}\;\left(\mu g\right)$	x	x	x	x	x	x	x	x
$\nabla_{front}\;\left(\mu g\right)$	x	x	x	x	x	x	x	x
$\nabla_{up}\;\left(\mu g\right)$	x	x	x	x	x	x	x	x
$\delta A_{right}\;{(}^{\prime })$	29.40	29.20	29.01	28.75	28.82	29.43	31.04	29.23
$\delta A_{front}\;{(}^{\prime })$	31.20	31.21	31.20	30.60	31.01	31.60	28.29	29.02
$\delta A_{up}\;{(}^{\prime })$	29.50	31.19	31.80	29.53	31.40	33.40	31.30	24.12
$\delta l_{right}\;\left(m\right)$	0.0113	0.0109	0.0107	0.0113	−0.202	0.011	0.103	−0.213
$\delta l_{front}\;\left(m\right)$	0.0124	0.0115	0.0123	0.0121	0.0120	−0.247	0.108	−0.230
$\delta l_{up}\;\left(m\right)$	0.0106	0.0115	0.0114	0.0114	0.0107	0.096	−0.227	−0.227

**Table 2 sensors-22-04691-t002:** The difference between the yaw angle obtained by the visual aid initial alignment algorithm and the corresponding turntable.

The Experimental Group Number	1	2	3	Average Value
Difference in yaw ${(}^{\circ })$	0.4438	0.5890	0.6106	0.5478

## Data Availability

Not applicable.

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
