# Peer review of "A Vision Aided Initial Alignment Method of Strapdown Inertial Navigation Systems in Polar Regions"

_sensors, 2022, doi:10.3390/s22134691_

Round 1
Reviewer 1 Report
In this manuscript, the authors claim “A vision aided initial alignment method of strapdown inertial navigation systems in polar regions.” the simulation results show that the proposed method can improve the accuracy of the triaxial alignment of SINS in polar regions. However, there are several points the authors need to address:
1) In the abstract, the authors claimed “…cannot self-align due to high altitudes”, why?
2) In the polar region, how to get the features of the external situations.
3) Why not use the GPS for initial alignment?
4) Since the vision results are relative information, how to use the vision results to calculate the absolute yaw?
5) It lacks the experimental tests; I recommend adding the experimental results to show the performance of the proposed method.
Author Response
reply to the question:
1) In the abstract, the authors claimed “…cannot self-align due to high altitudes”, why?
It is a clerical error in high altitude area, it should be in high latitude area. It has also been revised in the article.
The polar environment is very special. In the high-latitude areas, the Earth meridians converge at the poles quickly. The Strapdown inertial navigation system (SINS) of autonomous underwater vehicle (AUV) cannot achieve self-alignment at high latitudes[1].
2) In the polar region, how to get the features of the external situations.
Position and heading information can be obtained by starlight navigation, and attitude information can be obtained by accelerometer. The camera was used to measure the known position and attitude of the fixed plate to obtain the information of feature points. The velocity error and position error of strapdown inertial navigation system in horizontal coordinate system are deduced.
3) Why not use the GPS for initial alignment?
Ionospheric scintillation plays an important role in the accuracy, reliability and availability of GPS system. Ionospheric scintillation occurs at high latitudes. The strong phase change and amplitude fluctuation of the signal may interfere with the operation of the GPS receiver and have a great impact on the accuracy, reliability and availability of the GPS system. For example, in reference 2, the influence of different phase conditions in high latitude on GPS accuracy was studied[2]. Therefore, GPS cannot be used for initial alignment due to the unreliability of the polar GPS system.
4) Since the vision results are relative information, how to use the vision results to calculate the absolute yaw?
Through vision, the position and attitude of p frame at time T in the camera coordinate system can be obtained, and the known position and attitude of the fixed plate measured by the camera are used as prior information to obtain the transformation matrix from the camera coordinate system to the navigation coordinate system. Thus, the relative information of vision is converted to absolute yaw.
5) It lacks the experimental tests; I recommend adding the experimental results to show the performance of the proposed method.
Due to limited funds, the depolarizing zone test conditions are not available, and only simulation verification and analysis can be carried out. Through simulation experiments, eight kinds of oscillation modes, including triaxial rotation, biaxial rotation, uniaxial rotation and stationary, have been studied in high latitude region. The performance of this method is well verified by the simulation of various swing states.
[1]Anderson, E.W.I. Navigation in Polar Regions. J. Navig. 1957, 10, 156–161. [CrossRef]
[2]K. Meziane, A. Kashcheyev, P. T. Jayachandran and A. M. Hamza, "On the latitude-dependence of the GPS phase variation index in the polar region," 2020 IEEE International Conference on Wireless for Space and Extreme Environments (WiSEE), 2020, pp. 72-77, doi: 10.1109/WiSEE44079.2020.9262655.

Reviewer 2 Report
I would like to recommend a quick acceptance of this paper after the authors fixing some typo and grammatic errors.
If the authors can also state the accuracy of the initial alignment, that will be great.
I will also appreciate if the authors can justify the rationale of EKF rather than two stage adaptive EKF or other similar filters.
In addition to the simulation, do the authors have real data?
Author Response
reply to the question:
- I would like to recommend a quick acceptance of this paper after the authors fixing some typo and grammatic errors.
Some spelling and grammatical errors in the article have been corrected and revised, and the article has been thoroughly checked after the modification. Thank you for your advice.
- If the authors can also state the accuracy of the initial alignment, that will be great.
The performance and feasibility of the proposed method are verified by simulation experiments of three-axis rotation, two-axis rotation, single-axis rotation and static swing. The estimation accuracy of the celestial misalignment Angle is significantly higher than that of the single- axis roll, single-axis yaw and stationary rotation when the rotation is three-axis, two-axis and single axis pitching. Meanwhile, the residual installation error Angle between the camera and the INERTIAL navigation can be accurately estimated, which is consistent with the theoretical analysis. When the gyroscope is biased at 0.05°/h and the accelerometer is biased at 100μg, the triaxial alignment accuracy of the strapdown inertial navigation system is 0.11 ', 0.17 'and -0.30'. It has the same effect as the estimation of the deviation Angle in the case of uniaxial swing, biaxial swing and triaxial swing after alignment for 300s.
Table 1 Comparison between theory and practice of misalignment Angle under triaxial swing
|
|
theoretical value |
actual value |
|
0 |
0.11 |
|
|
0 |
-0.17 |
|
|
0 |
-0.30 |
- I will also appreciate if the authors can justify the rationale of EKF rather than two stage adaptive EKF or other similar filters.
Based on EKF filtering, this paper uses the camera to obtain the position information of feature points, deduces the speed error and position error of the system in the horizontal coordinate system, and updates the state and measurement. When AUV swing through the known feature point relative to the camera movement, computing the position and posture of the AUV itself. SINS is combined with visual measurement, the observation results of AUV's own motion are used as measurement information, and the EKF algorithm is used to update the state equation and measurement equation, so as to complete the high-precision initial alignment of SINS with visual assistance.
- In addition to the simulation, do the authors have real data?
Due to limited funds, the depolarizing zone test conditions are not available, and only simulation verification and analysis can be carried out. Through simulation experiments, eight kinds of oscillation modes, including triaxial rotation, biaxial rotation, uniaxial rotation and stationary, have been studied in high latitude region. The performance of this method is well verified by the simulation of various swing states.
